# Improving Input-label Mapping with Demonstration Replay for In-context Learning

**Zhuocheng Gong**[1][*], **Jiahao Liu**[2], **Qifan Wang**[3]
**Jingang Wang**[2], **Xunliang Cai**[2], **Dongyan Zhao**[1,4,5][†], **Rui Yan**[6][†]
[1]Wangxuan Institute of Computer Technology, Peking University
[2]Meituan; [3]Meta AI; [4]National Key Laboratory of General Artificial Intelligence
[5]Beijing Institute for General Artificial Intelligence
[6]Gaoling School of Artificial Intelligence, Renmin University of China
{gzhch,zhaody}@pku.edu.cn, ruiyan@ruc.edu.cn, wqfcr@fb.com
{liujiahao12,wangjingang02,caixunliang}@meituan.com

## Abstract

In-context learning (ICL) is an emerging capability of large autoregressive language models where a few input-label demonstrations are appended to the input to enhance the model's understanding of downstream NLP tasks, without directly adjusting the model parameters. The effectiveness of ICL can be attributed to the strong language modeling capabilities of large language models (LLMs), which enable them to learn the mapping between input and labels based on in-context demonstrations. Despite achieving promising results, the causal nature of language modeling in ICL restricts the attention to be backward only, i.e., a token only attends to its previous tokens, failing to capture the full input-label information and limiting the model's performance. In this paper, we propose a novel ICL method called Repeated Demonstration with Sliding Causal Attention, (RDSCA). Specifically, we duplicate later demonstrations and concatenate them to the front, allowing the model to 'observe' the later information even under the causal restriction. Besides, we introduce sliding causal attention, which customizes causal attention to avoid information leakage. Experimental results show that our method significantly improves the input-label mapping in ICL demonstrations. We also conduct an in-depth analysis of how to customize the causal attention without training, which has been an unexplored area in previous research.

## 1 Introduction

Large language models (LLMs) have become the backbone of various natural language processing tasks in different fields. One of the most remarkable abilities of LLMs is in-context learning (ICL) (Brown et al., 2020). By providing a few demonstrations and instructions into the input, along with the input queries, LLMs can perform well in new tasks without requiring fine-tuning. The secret of ICL is to formulate the input as the natural language generation task, then the LLM can be activated to prompt knowledge learned in the pre-training stage.

Despite the promising results demonstrated by existing ICL models (Gonen et al., 2022; Wei et al., 2022), their causal nature in language modeling restricts each token's attention solely to its preceding tokens. As a result, these models fail to capture the complete input-label information, thereby limiting their overall performance. Specifically, the pre-training objective of the current autoregressive LLMs focuses on predicting future tokens based on past ones (Radford et al., 2018), implemented with causal attention. While this approach works well for modeling regular sequences, it becomes less effective when applied to ICL tasks. The limitation of causal attention restricts ICL demonstrations to having only left context, which hampers the model's ability to fully comprehend and exploit the input-label relationship.

Unlike tokens in a sentence that possess sequential dependencies, there is no inherent sequential relationship between the demonstrations in the ICL input. Therefore, it is desirable for these demonstrations to interact with one another comprehensively, rather than relying solely on later demonstrations attending to earlier ones, while the reverse is not possible. Intuitively, if we can enable each demonstration to attend to all the others, we can potentially obtain a more sophisticated context for the

---

[*]Work done during an internship at Meituan.
[†]Corresponding authors: Dongyan Zhao (zhaody@pku.edu.cn) and Rui Yan (ruiyan@ruc.edu.cn).

ICL query. However, achieving this on an LLM pre-trained with the objective of causal language modeling is not straightforward. Simply removing the causal restriction and allowing the model to have access to the right context during inference is not feasible, as it would result in a significant disparity between training and inference conditions.

In this work, we focus on capturing full input-label mapping information from demonstrations. To achieve this target, we propose two techniques. The first is **Repeated Demonstration**, where we replicate later demonstrations and concatenate them to the front. This allows the model to 'observe' the later information even under the causal restriction. For example, if we consider four demonstrations represented by $d_1d_2d_3d_4$, the input sequence after replication becomes $d'_2d'_3d'_4d_1d_2d_3d_4$. However, simply duplicating demonstrations brings about a new problem: we do not want to attend a demonstration twice, as this may cause the model to take shortcuts by learning to repeat the answer of its first encounter, rather than learning the input-label mapping. To address this, we propose the second technique, the **Sliding Causal Attention**, which customizes the original causal attention by restricting the attention window so that each demonstration can only attend to all other demonstrations once. In the case of four demonstrations, attention windows are $d'_2d'_3d'_4d_1$, $d'_3d'_4d_1d_2$, $d'_4d_1d_2d_3$, and $d_1d_2d_3d_4$, respectively. Through experiments, we demonstrate that our proposed method (Repeated Demonstrations with Sliding Causal Attention, RDSCA) significantly enhances the ability to learn input-label mapping from ICL demonstrations.

Our proposed sliding casual attention is the first attempt that customizes the causal attention in the inference stage without further training. We investigate a number of different designs for customizing causal attention and reach some principles for the success of ICL. For example, we find that the first <SOS> token plays an essential role. It should always be available to attend to no matter where the attention window slides. Besides, the size of the attention window determines the richness of the semantic context, thus it affects the performance greatly. Our contributions are summarized as:

- To the best of our knowledge, we are the first to identify the limitation of causal language modeling in the ICL and to introduce a novel approach for enabling effective interactions

between demonstrations.

- We validate the feasibility of customizing causal attention during the inference stage without further training and conduct further analysis on causal attention customization. We believe this idea has great potential and sheds new light on optimizing ICL and other large model inference scenarios.

- We conduct experiments on several text classification datasets to evaluate the effectiveness of our proposed method. The experimental results clearly demonstrate that our approach significantly enhances the input-label mapping in ICL demonstrations.

## 2 Backgrounds

### 2.1 Causal Language Modeling

Most of the current decoder-only LLMs employ causal language modeling as the pre-training objective (Radford et al., 2018), which aims to predict future tokens based on the past.

$$\arg\min_{\theta} \sum_i P_\theta(u_i|u_0, u_1, ..., u_{i-1}) \qquad (1)$$

In causal language modeling, the model only attends to tokens that occur before (the left context), resulting in a unidirectional attention scheme known as **causal attention**. This approach enables the model to process each token in the input sequence in order, without accessing any information from the future (the right context).

**Causal Attention Mask**   Practically, a causal attention mask is used to implement causal attention, which guarantees unidirectionality by masking all right-to-left attention connections and only allowing right-to-left connections. Formally, the attention mask is a binary-valued matrix $M \in \{0,1\}^{n \times n}$, where $n$ is the total sequence length. The element $m_{ij}$ in $M$ indicates whether the $j$-th token in the sequence can attend to the $i$-th token, with a value of $1$ for yes and $0$ for no. Therefore, the causal attention mask matrix is a lower triangular matrix where $m_{ij} = 0, \forall i < j$.

### 2.2 In-context Learning

Now we formally introduce the definition and basic notations of in-context learning. We focus on classification tasks and causal language models. Given

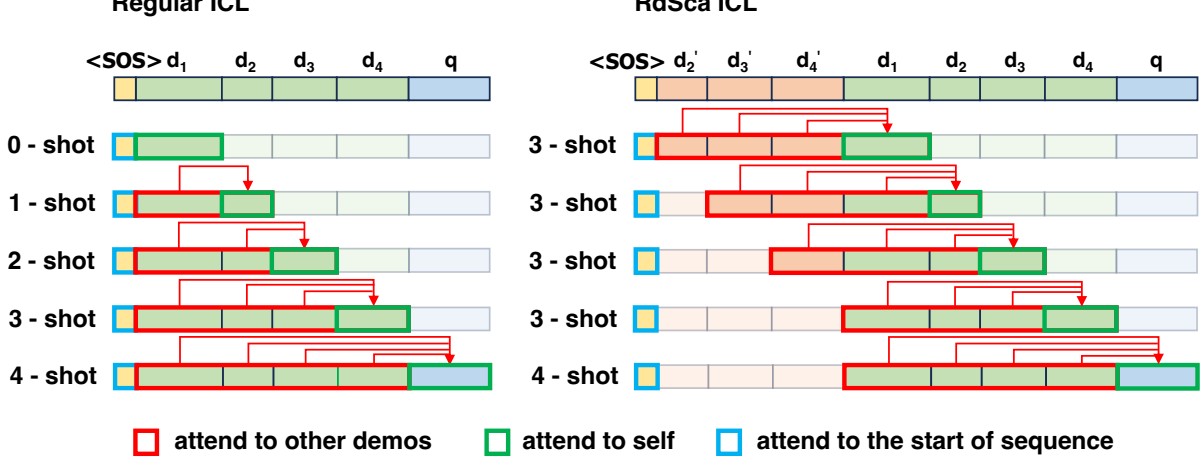

Figure 1: An illustration for regular ICL and our RDSCA ICL. Regular ICL uses vanilla causal attention, causing the uneven interaction between demonstrations. In RDSCA, each demonstration can attend to all the others thanks to demonstration repetition and sliding causal attention.

$K$ input-label pairs $\{x_i, y_i\}_{i=1}^K$ and a query $x_q$, the objective is to predict the label of the query $y_q$ by predicting the next token. Formally,

$$\arg\min_{y_q \in \mathcal{C}} P(y_q|(x_1, y_1), (x_2, y_2), ..., (x_K, y_K), x_q)$$
(2)

where $\mathcal{C}$ is the label set.

To perform ICL successfully, we transform the classification task into a natural language generation task by adding templates to the query and demonstrations. Additionally, discrete labels are mapped to label words, such as "positive" and "negative" for sentiment classification. We denote the demonstration with templates and label words as $d_i = T(x_i, y_i)$. The entire input to the model is formed by concatenating the demonstrations and the query $q = T(x_q, \_)$, resulting in $d_1 d_2 ... d_k q$.

**Semantically-unrelated label ICL (SUL-ICL)** In regular ICL, natural language words that are closely related to the task objective are used as label words. This allows the model to utilize semantic prior knowledge. For example, when conducting sentiment classification, label words such as "positive" and "negative" are highly semantically related to the sentiment labels of the input. During pretraining, the model is exposed to similar patterns and can learn to associate these label words with the corresponding sentiment labels. In this paper, we eliminate the contribution of semantic priors and perform semantically-unrelated label ICL (Wei et al., 2023). In this setting, natural language labels are replaced with semantically-unrelated labels. This approach forces the model to rely solely on input-label mappings to perform ICL.

## 3 Method

### 3.1 Defect of Traditional ICL

During few-shot ICL, we combine the labeled demonstrations with the query by concatenating them and feeding them into the LLM. The labeled demonstrations provide valuable in-context input-label information, which is included in the few-shot ICL input. As a result, few-shot ICL consistently achieves better performance compared to zero-shot ICL.

However, few-shot ICL under the current scheme of causal language modeling has a defect. Due to the restriction of causal attention, each demonstration only has access to half of the full context (*i.e.*, the left context). As a result, it cannot 'observe' demonstrations that follow it. For example, in 4-shot ICL, if we consider the third demonstration as the last query (which makes sense because tokens that come after it have no influence on it), then the first two demonstrations serve as its context while the fourth demonstration is ignored. In this sense, we can regard predicting the label of the first demonstration as a zero-shot ICL task, the second as a one-shot ICL, the third as a two-shot ICL, and so on, as shown in Figure 1.

While restricting the model from accessing the right context makes sense when modeling regular natural language sequences to prevent information leakage, it is unnecessary for few-shot ICL since there is no dependency between demonstrations. The causal restriction limits earlier demonstrations

from accessing later information, resulting in only half of the available information being utilized.

## 3.2 Repeated Demonstrations

Enabling each demonstration to 'observe' later demonstrations is not a trivial task. Simply replacing causal attention with full attention to allow the model to receive both left and right context is not feasible, as the model is pre-trained with causal language modeling. Switching to full attention during inference would result in a significant loss in performance, as we have verified through experiments (Section 4.4). Therefore, we propose the **Repeated Demonstration** method to establish sufficient interactions among demonstrations while still adhering to the premise of causal attention. The method is based on a simple idea: duplicating all the other demonstrations except the first one and concatenating them to the front of the sequence. By doing so, we expand the input sequence $d_1 d_2 ... d_K q$ to $d'_2 ... d'_K d_1 d_2 ... d_K q$ where $d'_i$ is the duplication of $d_i$. This operation allows each demonstration to attend to all the others by adding the later demonstrations to its left context. As a result, the model can make full use of demonstrations while still obeying the causal attention restriction.

## 3.3 Sliding Causal Attention

The Repeated Demonstration method alone is not sufficient to achieve our target, as it introduces a new problem of duplicated information in the context. Under causal attention, a token can attend to all tokens that precede it. When combined with repeated demonstrations, this can result in the same information being duplicated in the context. For example, when performing 4-shot ICL with repeated demonstrations, the input is $d'_2 d'_3 d'_4 d_1 d_2 d_3 d_4 q$. The context of $d_3$ is $d'_2 d'_3 d'_4 d_1 d_2 d_3$ where some demonstrations appear twice. Repetitive information can cause the model to learn a shortcut to predict the label by repeating the label of the same demonstration that appeared for the first time, rather than learning the input-label mapping as expected. We will provide a detailed explanation of this phenomenon in the experimental section (Section 4.4).

To tackle this issue, we propose **Sliding Causal Attention**, which utilizes a sliding attention window to limit the range of tokens that each demonstration can attend to. This approach effectively prevents the occurrence of repetitive information and ensures that each demonstration has a non-

repetitive context. Specifically, the attention window is defined as follows:

$$window(x) = \begin{cases} d'_{i+1} ... d'_K d_1 ... d_i & x = d_i, i \leq K \\ d_1 d_2 ... d_K q & x = q \end{cases}$$

(3)

Then we explain how sliding causal attention works. We use the window size $W$ to represent the number of demonstrations contained within it. In our main setting, we use $W = K$, indicating each attention window contains all the demonstrations. As the window slides, a new demonstration enters while an old one exits. This ensures that there are always consistent $K$ different demonstrations within the context.

Additionally, we find that the `<SOS>` token, which represents the first token of the sequence, is crucial to the model's performance. Therefore, we add `<SOS>` to every attention window to ensure that each token can attend to `<SOS>`. We will provide further explanations on this in the experimental section 4.4.

# 4 Experiments

## 4.1 Setup

**Models**  We conducted experiments on decoder-only causal language models. We utilized the LLAMA (Touvron et al., 2023) model family with varying scales, including 7B, 13B, 30B, and 65B.

**Tasks**  We evaluate on classification tasks, including **SST-2** (Socher et al., 2013), **CB** (De Marneffe et al., 2019), **RTE** (Dagan et al., 2005; Wang et al., 2019a), **AGNews** (Zhang et al., 2015), **QQP** (Wang et al., 2019b), and **QNLI** (Wang et al., 2019b). If the dataset includes a validation split, we evaluate the model's performance on the validation set. Otherwise, we evaluate on the test set. Datasets are obtained from Huggingface `Datasets` library[*].

**Other Details**  For all experiments, we use $K = 4$ demonstrations by default. Demonstrations are uniformly sampled from the training data. We utilize prompt templates from `PromptSource`[†] (Bach et al., 2022). For each dataset, we use four different templates and select a set of $K$ training examples using 4 different random seeds. Therefore, the reported results are the average of 16 different runs. We would like to emphasize that we run all methods using the same random seeds, ensuring that

---

[*]https://huggingface.co/docs/datasets/index
[†]https://github.com/bigscience-workshop/promptsource

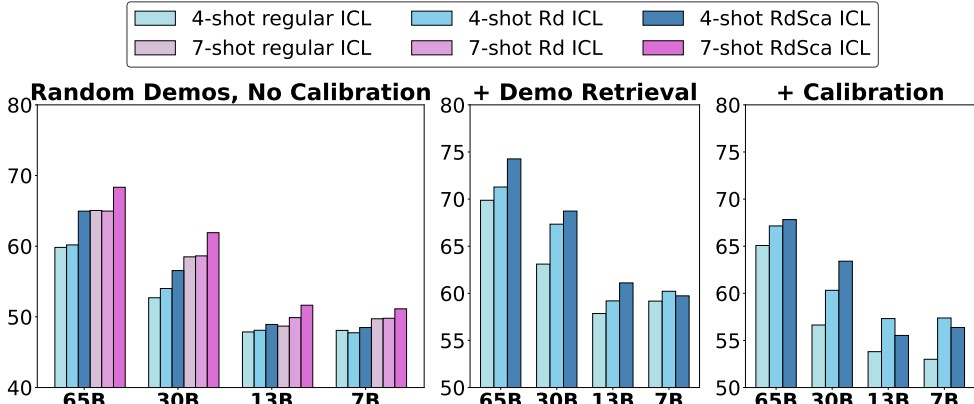

Figure 2: Averaging performance of RDSCA on LLAMA of different scales. **Left**: ICL with random-selected demonstrations. We report the results of both 4-shot and 7-shot ICL for this setting. **Middle**: We combine our method with the demonstration retrieval technique. We employ KATE (Liu et al., 2022) to retrieve target demonstrations. **Right**: we combine with PROTOTYPICAL CALIBRATION (Han et al., 2022) to further improve performance. Full results can be found in Appendix A

the demonstrations used by each method are identical. This was done to eliminate any potential bias caused by differences in the demonstrations seen by each method and ensure a fair comparison. Due to limited computing resources, we evaluate a random sample of 200 examples for tasks with a large validation set, instead of the entire dataset. We have found that this sample size is sufficient to obtain stable and reliable results.

## 4.2 Main Results

Figure 2 presents a comparison between our method and regular ICL. We observe a clear correlation between the performance of semantically unrelated label ICL and the model scale: increasing the model scale improves performance for all methods. In the case of LLAMA-7B, the performance is poor, and the performance gap between different methods is small, indicating that it is challenging for small models to learn the input-label mapping from demonstrations while the semantic meaning of labels is removed. As the model size scales up to 30B and 65B, the models begin to exhibit strong ICL capabilities, and the performance of different methods varies. This phenomenon is consistent with the findings of Wei et al. (2023), which suggest that larger language models learn the input-label mapping differently because the input-label mapping learning ability only emerges when the model is large enough. In comparison with ICL with random demonstrations, we find that retrieving demonstrations that are more relevant to the query can significantly improve the performance

on all scales. Furthermore, calibration has a positive influence on the ICL results, showing that our method can be combined with other techniques to further enhance the overall performance.

**RDSCA improves input-label mapping** We observe a significant performance boost over regular ICL in both 4-shot and 7-shot settings on LLAMA-30B and LLAMA-13B. Our method shows an average improvement of 8.4% on LLAMA-30B and 10.5% on LLAMA-65B compared to regular ICL, indicating a significant advantage in learning the input-label mapping from demonstrations. As mentioned in previous sections, we believe that in the scheme of causal language modeling, regular ICL only utilizes half of the context (left context) for the demonstrations. In contrast, our method makes use of all the available information, which enhances the representations of demonstrations and provides richer semantics for predicting the query's target.

## 4.3 Results on MMLU

|  | REGULAR ICL | RDSCA |
|---|---|---|
| humanities | 65.99 | 68.44 |
| STEM | 51.84 | 51.57 |
| social sciences | 72.29 | 73.60 |
| other | 63.34 | 64.65 |
| average | 62.20 | 63.66 |

Table 1: LLAMA-65B 4-shot ICL results on MMLU.

We also validate LLAMA-65B on

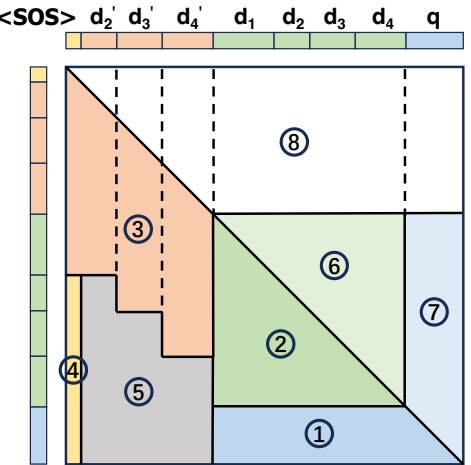

Figure 3: Illustration of the attention mask, which we divide into multiple regions and label with numbers. The specific meanings are as follows: ① query-to-demonstrations attention; ② inter-demonstrations right-to-left attention; ③ inter-demonstrations(repeat) right-to-left attention; ④ attention to the start-of-sentence token; ⑤ attention to redundant context; ⑥ inter-demonstrations left-to-right attention;⑦ demonstrations-to-query attention;⑧ other left-to-right attention.

MMLU (Hendrycks et al., 2021), which is a more advanced benchmark for evaluating the abilities of LLMs. The results are shown in Table 1.

### 4.4 Looking Deeper into Causal Attention Customization

In this section, we conduct a further investigation to look deeper into the effectiveness of sliding causal attention. *How and why does sliding causal attention work and are there alternatives when customizing causal attention?* To answer these questions, we implement the following ablations:

- FULL ATTN. This uses full attention instead of causal attention so the model can attend to both the left and right context.

- RD. This is the ablation of RDSCA that uses original causal attention without sliding causal attention.

- RDSCA w/o <SOS>. In this ablation of RD-SCA, when the attention window moves forward, latter tokens cannot attend to the beginning of the sentence, *i.e.*, the <SOS> token.

Figure 3 shows how we divide the whole attention matrix into various regions. The above-mentioned methods have unique access to these regions. Details and experimental results are shown in Table 2.

**Can we break the causal restriction?** First, we are concerned about the limitations of causal attention. Intuitively, if we can remove the causal restriction during inference by using non-causal attention and providing the model with the full context, we can fully utilize the demonstration information. However, our experimental results indicate that this simple idea is not feasible. The performance of FULL ATTN. is no better than random guessing, but why? There is a simple explanation: the gap between causal attention training and non-causal attention inference is too huge. Since the model is trained using causal language modeling, it has no inductive bias toward data with the correct context. Therefore, using full attention during inference is out-of-distribution for a causal model.

**Sliding attention window matters.** Based on the results, it is evident that RD has little improvement over regular ICL. Although RD provides richer interaction between demonstrations than regular ICL, the lack of attention restrictions causes the same demonstration to attend to its first appearance. Consequently, the model takes a shortcut and repeats the answer of the first appearance instead of learning the input-label mapping to predict the result. Therefore, the improvement brought by simply repeating the demonstrations is limited. However, the situation changes when we add the sliding causal window, which blocks repeated information and forces the demonstrations to attend only to other unseen demonstrations, ensuring that there is no information leakage. According to the evaluation results, this approach successfully enables the model to better capture the input-label mapping information, as expected.

This ablation study further indicates another insight: the default causal attention may not be always optimal in ICL. Previous research aimed at improving ICL has mainly focused on template construction and demonstration selection, with little attention paid to the causal attention masks. Our study shows that customizing attention masks can be a new technique for enhancing ICL, which is worthy of further research.

**Attending to the <SOS> token is essential.** Next, we examine the role of the first token <SOS> in causal attention customization. If not treating the <SOS> token separately, as the attention window moves, the first token slides out of the scope of the attention window. This means that the latter demon-

| Methods | Attention Regions | Is causal? | LLAMA-65B | LLAMA-30B |
|---|---|---|---|---|
| RANDOM GUESS | - | - | 43.06 | 43.06 |
| REGULAR ICL | ①+② | Yes | 59.83 (+16.77) | 52.71 (+9.65) |
| FULL ATTN. (v1) | ①+②+⑥ | No | 45.15 (+2.09) | 47.84 (+4.78) |
| FULL ATTN. (v2) | ①+②+⑥+⑦ | No | 44.32 (+1.26) | 47.36 (+4.30) |
| RD | ①+②+③+④+⑤ | Yes | 60.18 (+17.12) | 54.02 (+10.96) |
| RDSCA w/o <SOS> | ①+②+③ | Yes | 32.20 (−10.86) | 12.96 (−30.10) |
| RDSCA | ①+②+③+④ | Yes | 64.96 (+**21.90**) | 56.55 (+**13.49**) |

Table 2: Ablations on causal attention customization. We report the averaging performance on all tasks. We implement two versions of the FULL ATTN., which differ in whether including left-to-right attention of the query.

strations and the query cannot attend to <SOS>. Our experiments show that in this setting the ICL ability is severely affected (①+②+③ in Table 2). However, when we manually add the <SOS> token to the attention window (as in the case of RD-SCA), we observe a significant improvement in performance. This comparison demonstrates that the <SOS> token is crucial for the model to perform correctly. But why does such a seemingly insignificant token have such a significant impact on performance? We believe that the significance of the <SOS> token lies in allowing the model to treat the tokens under the attention window as a valid sequence rather than a fragment of the whole sequence.

As the model is pre-trained on sequences that all start with <SOS>, it does not know how to handle sequences that do not start with <SOS>. Therefore, when customizing causal attention, we need to make sure that the model regards the tokens in the attention window as a valid sequence starting with <SOS>. Only in this way, we can activate the model's language modeling ability correctly.

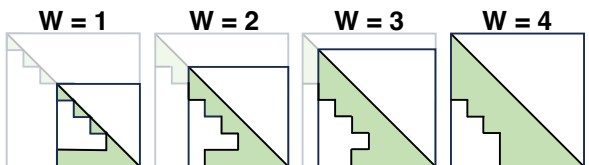

Figure 4: Visualization of attention masks for different window sizes.

## 4.5 Ablation on Attention Window Size

In this section, we discuss the impact of window size. We denote the attention window size of the original RDSCA as $W = K$, meaning that there

| Window | n-shots | LLAMA-65B | LLAMA-30B |
|---|---|---|---|
| REGULAR | 0-1-2-3-4 | 69.87 | 63.10 |
| W=1 | 0-0-0-0-4 | 60.83 | 52.78 |
| W=2 | 1-1-1-1-4 | 71.33 | 60.83 |
| W=3 | 2-2-2-2-4 | 74.01 | 65.70 |
| W=4 | 3-3-3-3-4 | 74.27 | 68.73 |
| W=4 | 3-3-3-3-3 | 70.99 | 64.44 |

Table 3: Average performance on all tasks for different attention window sizes. We also report the number of demonstrations contained in the context of each demonstration. For example, "0-1-2-3-4" means that, for regular ICL, the first demonstration has no context, the second sees one demonstration, the third sees two, and so on. In this experiment, we use KATE to retrieve demonstrations instead of random-selected ones. In this way, we save computational resources because there is no need to run experiments over different random seeds.

are $K$ demonstrations in each window. In this case, each demonstration can attend to all other ones, so each demonstration can be considered as performing $(K − 1)$-shot ICL. As shown in Figure 4, we employ smaller window sizes $W$ on RDSCA and see what happens to the ICL ability. Intuitively, reducing the size of the sliding window means reducing the context (the number of demonstrations) that the current one is able to attend to. Thus the model's ability to learn the input-label mapping from context would be affected. Especially, when $W = 1$, each demonstration can only attend to itself, which is equivalent to zero-shot ICL. The results are shown in Table 3. As expected, the window size is highly correlated with ICL performance: the larger the window size, the better the performance. Surprisingly, we notice that when $W = 2$, i.e., one-shot ICL for all demonstrations, RDSCA is already comparable with regular ICL on

both 30B and 65B LLAMA. This indicates that our ICL method makes more efficient exploitation of the demonstrations in ICL. The last row of Table 3 shows when the query not seeing all 4 demonstrations, the performance drops.

### 4.6 What If Adding More Demonstrations?

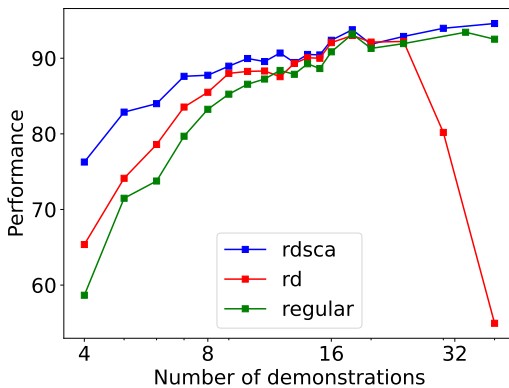

Figure 5: Increasing the number of demonstrations for LLAMA-65B on SST-2.

In this section, we investigate how our method performs with more demonstrations. We start from $K = 4$ demonstrations and gradually add more. As shown in Figure 5, all methods receive significant performance gains when the number of demonstrations is relatively small. However, as the number of demonstrations increases, the benefits of adding more demonstrations gradually reduce. The ICL performance reaches a plateau at around $K = 16$. Before reaching this plateau, RDSCA consistently outperforms the vanilla-repeat and regular ICL, indicating that our method is robust to the increase in the number of demonstrations. However, this advantage gradually diminishes when adding more demonstrations. All methods seem to perform comparably at around $K = 16$, suggesting that the ability to learn from context is close to its upper bound at this point.

In addition, we observe an interesting phenomenon in the experiment. We find that the performance of RD sharply declines when there are too many demonstrations (decreasing by 12.02 points from $K = 24$ to 30, and 25.24 points from $K = 30$ to 40.), while RDSCA does not suffer such huge losses. We explain this phenomenon from two aspects: First, as mentioned earlier, if not customizing the causal attention, the model learns shortcuts from repeated examples rather than input-label

mapping, which leads to a decline in ICL performance. Second, due to the repetition of demonstrations, the context becomes too long when adding too many demonstrations, and some studies have shown that LLMs still have difficulty modeling long sequences. Our method can effectively solve these problems by customizing attention. Therefore, we believe that our exploration of customizing causal attention highlights a possible solution for tackling the long sequence modeling issue, which is worth further research in the future.

### 4.7 Discussions on Efficiency Issues

As repeating demonstrations involves expanding the sequence lengths, the model is required to process more tokens when compared with regular ICL, which may lead to computational inefficiency and extra memory consumption. However, we argue that these efficiency issues can be neglected practically. Thanks to the auto-regressive nature of Large Language Models LLMs, the representations of previous tokens are not dependent on later tokens. Therefore, in the inference phase, the representations of demonstrations can be pre-computed and stored offline, thereby mitigating the added computational burden. Moreover, our adaptation of the attention mask allows the LLM to focus only on tokens within the sliding window for key-value (KV) caching, rather than retaining information on all previous tokens. This optimization reduces the memory requirements of RdSca. Consequently, the memory consumption of RdSca remains on par with that of a standard ICL setup.

## 5 Related Works

### 5.1 In-context Learning

In-context learning (ICL) is an effective approach for adapting pre-trained LLMs to downstream tasks (Brown et al., 2020). This is achieved by adding task-specific templates and demonstrations before the test query, without updating model parameters. Recent works focus on enhancing ICL with various techniques. For example, Liu et al. (2022); Rubin et al. (2022); Ram et al. (2023); Luo et al. (2023) propose that choosing demonstrations more carefully results in better ICL performance. Some studies try improving ICL with more sophisticated prompt templates, either by hand-craft or by automation (Gonen et al., 2022). Some propose chain-of-thoughts (CoT) to elicit the reasoning abilities of LLMs by augmenting each demon-

stration with a chain of reasoning steps (Wei et al., 2022). Subsequent studies have continued to build upon this approach, achieving further improvements (Wang et al., 2022). Some studies explore the mechanism of ICL. Min et al. (2022) shows that randomly replacing labels in the demonstrations barely hurts performance so ground truth demonstrations are not required. (Wei et al., 2023) claims that the success of ICL relies on both semantic prior knowledge and input-label mappings. They find that larger models can better capture input-label mappings from demonstrations.

## 5.2 Attention Customization

There has been a number of work done on attention customization since Transformer was proposed (Vaswani et al., 2017). many of them focus on modeling long sequences (Kitaev et al., 2020; Beltagy et al., 2020) or improving efficiency through sparse attention (Tay et al., 2023). Some works have explored customizing attention to block certain dependencies. For example, Mu et al. (2023) prevents later tokens from attending to the prompt to compress context. These attention customizations can be viewed as some kind of modifications to the model architecture, which requires additional training. However, our study is the first to investigate the possibility of attention customization during inference without further training.

## 6 Conclusion

In this study, we introduce RDSCA, a novel ICL framework that enhances the learning of the input-label mapping from demonstrations. We propose that the causal attention mechanism of decoder-only LLMs restricts the model from fully exploiting the input-label mapping from demonstrations. To address this, we suggest repeating demonstrations to allow each demonstration to have full context and customizing the vanilla causal attention mechanism to prevent information leakage. Experimental results show that our method consistently improves the input-label mapping ability of ICL on LLMs of different scales. Furthermore, we delve deeper into causal attention customization and show how different attention settings affect ICL performance. Additionally, this work is the first to customize the causal attention of a pre-trained autoregressive LLM without further training, which may pave the way for further research in this direction.

## Limitations

Although RDSCA has demonstrated outstanding performance in capturing the full input-label mapping from demonstrations, there are some limitations to our work in this section that we must acknowledge. First, our method expands the sequence length by duplicating demonstrations, which leads to increased computation costs and inference latency. This may limit its practicality in computation-constrained scenarios. Moreover, we have not evaluated our method on more complex tasks, nor have we determined its performance with chain-of-thought scenarios. Therefore, in the future, we need to investigate how it can be extended to a broader range of task scenarios.

## Acknowledgments

This work is supported by National Key R&D Program of China (No. 2021YFC3340304) and National Natural Science Foundation of China (NSFC Grant No. 62122089). Jingang Wang is funded by Beijing Nova Program (Grant No. 20220484098). We sincerely thank all reviewers for their valuable comments and suggestions, which are crucial for improving our work.

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

# A Full results

| Methods | 65B | 30B | 13B | 7B |
|---|---|---|---|---|
| REGULAR ICL | 59.82 | 55.18 | 47.79 | 49.72 |
| RD | 63.66 | 59.91 | 49.68 | 49.34 |
| RDSCA | 75.76 | 69.32 | 53.03 | 50.92 |
| *with demonstration retrieval* | | | | |
| REGULAR ICL | 74.75 | 65.62 | 58.00 | 56.88 |
| RD | 77.13 | 73.63 | 56.38 | 59.25 |
| RDSCA | 87.38 | 82.13 | 68.88 | 64.63 |

Table 4: Full evaluation results on SST-2

| Methods | 65B | 30B | 13B | 7B |
|---|---|---|---|---|
| REGULAR ICL | 76.79 | 58.15 | 44.98 | 50.67 |
| RD | 76.12 | 56.49 | 42.94 | 48.34 |
| RDSCA | 76.90 | 56.17 | 46.97 | 50.66 |
| *with demonstration retrieval* | | | | |
| REGULAR ICL | 78.12 | 64.73 | 45.54 | 57.59 |
| RD | 79.72 | 60.38 | 49.06 | 59.91 |
| RDSCA | 79.72 | 60.38 | 52.83 | 57.08 |

Table 5: Full evaluation results on CB

| Methods | 65B | 30B | 13B | 7B |
|---|---|---|---|---|
| REGULAR ICL | 53.67 | 49.13 | 61.50 | 59.62 |
| RD | 53.20 | 53.37 | 62.54 | 57.96 |
| RDSCA | 59.68 | 53.44 | 60.05 | 57.23 |
| *with demonstration retrieval* | | | | |
| REGULAR ICL | 60.17 | 53.85 | 67.94 | 63.47 |
| RD | 61.79 | 62.95 | 70.38 | 64.20 |
| RDSCA | 64.87 | 61.54 | 66.79 | 64.36 |

Table 6: Full evaluation results on QQP

| Methods | 65B | 30B | 13B | 7B |
|---|---|---|---|---|
| REGULAR ICL | 59.38 | 53.45 | 47.84 | 46.56 |
| RD | 58.27 | 52.28 | 47.77 | 47.63 |
| RDSCA | 60.51 | 54.49 | 45.33 | 47.29 |
| *with demonstration retrieval* | | | | |
| REGULAR ICL | 63.21 | 60.90 | 51.79 | 52.56 |
| RD | 62.03 | 65.67 | 51.66 | 53.14 |
| RDSCA | 64.43 | 66.15 | 51.27 | 52.98 |

Table 7: Full evaluation results on QNLI

| Methods | 65B | 30B | 13B | 7B |
|---|---|---|---|---|
| REGULAR ICL | 36.97 | 36.33 | 29.69 | 28.19 |
| RD | 37.86 | 36.49 | 30.31 | 27.64 |
| RDSCA | 43.85 | 40.09 | 31.12 | 30.89 |
| *with demonstration retrieval* | | | | |
| REGULAR ICL | 70.75 | 70.38 | 67.12 | 67.50 |
| RD | 74.63 | 76.12 | 72.00 | 74.12 |
| RDSCA | 75.13 | 76.25 | 70.12 | 70.75 |

Table 8: Full evaluation results on AG-News

| Methods | 65B | 30B | 13B | 7B |
|---|---|---|---|---|
| REGULAR ICL | 72.36 | 63.99 | 55.4 | 53.76 |
| RD | 72.0 | 65.56 | 55.4 | 55.6 |
| RDSCA | 73.07 | 65.8 | 57.07 | 53.95 |
| *with demonstration retrieval* | | | | |
| REGULAR ICL | 72.25 | 63.12 | 56.75 | 52.75 |
| RD | 72.39 | 65.27 | 55.7 | 54.66 |
| RDSCA | 74.07 | 65.92 | 56.74 | 53.23 |

Table 9: Full evaluation results on RTE