# OpenReview forum: "Improving Input-label Mapping with Demonstration Replay for In-context Learning"
_EMNLP/2023/Conference — EMNLP 2023 Findings_

### Official Review · Reviewer_6837 · 2023-07-30

**Soundness:** 4

**Excitement:**

4: Strong: This paper deepens the understanding of some phenomenon or lowers the barriers to an existing research direction.

**Paper Topic And Main Contributions:**

After rebuttal: The authors effectively addressed my comments.

------------------

This paper proposes a new mechanism to improve in-context learning (ICL) by adding repeat demonstrations and sliding window attention, to enable each demonstration to attend to all the others and eliminate sequential dependencies.
This work uses LLaMA (7B ~ 65B) models and validates the proposed approach on some "classic" text classification datasets (SST-2, CB, RTE...).

**Reasons To Accept:**

1. The motivation is clear, and the proposed approach is intuitive.
2. Significant improvement is demonstrated on used benchmarks.

**Reasons To Reject:**

1. On the one hand, the work shows an interesting result: you can boost performance by duplicating 4-shots and manipulating the attention. On the other hand, 4-shots RdSca ICL cannot outperform regular 7-shots, according to Figure 2, which suggests that using regular ICL might still be the best choice with the same amount of computation budget.
2. The selected benchmark seems a bit outdated. Just like people use MMLU/BigBench to evaluate LLM instead of SST/CB, Such LLM-based ICL approaches need to be validated on MMLU/BigBench to be more convincing.

**Reproducibility:**

4: Could mostly reproduce the results, but there may be some variation because of sample variance or minor variations in their interpretation of the protocol or method.

**Reviewer Confidence:**

4: Quite sure. I tried to check the important points carefully. It's unlikely, though conceivable, that I missed something that should affect my ratings.

---

> ### Author Rebuttal · Authors · 2023-08-27
>
> Thank you for your constructive comments and suggestions. They are exceedingly helpful in improving our paper.
>
> #### **Regarding "4-shots RdSca ICL cannot outperform regular 7-shots, which suggests that using regular ICL might still be the best choice with the same amount of computation budget"**
>
> Thanks for this insightful question. We agree with the reviewer that regular ICL with more demonstrations can outperform RdSca with fewer demonstrations. In the following, we provide several explanations to further clarify this.
>
> - It's a common observation that ICL tends to exhibit improved performance as more demonstrations are added, whether in the case of regular ICL or RdSca ICL. However, the primary advantage of RdSca lies in its ability to maximize the utility of a fixed number of ICL demonstrations. This capability becomes particularly valuable when dealing with situations where data is extremely limited. MMLU serves as an illustrative example, given that it contains only 5 demonstrations in the development set. Consequently, attempting a 7-shot ICL approach on MMLU is impractical.
> - Regarding the additional computational resources required by RdSca, we assert that these demands are negligible since LLM inference tends to be limited by memory-IO rather than compute power [1]. Furthermore, in practical, large-scale ICL inference scenarios, it's possible to pre-compute and store the representations of demonstrations offline, leveraging the auto-regressive nature of LLMs. Therefore, the incremental computation overhead is not a significant concern.
> - From a memory consumption perspective, 4-shot RdSca is as efficient as 4-shot regular ICL, and far less consumptive than 7-shot regular ICL because RdSca only needs to track the status of at most 4 demonstrations for KV-caching thanks to the sliding window mechanism.
>
> We will provide further clarification during revision.
>
> [1] Shazeer N. Fast transformer decoding: One write-head is all you need[J]. arXiv preprint arXiv:1911.02150, 2019.
>
> #### **Regarding more advanced benchmark**
> > The selected benchmark seems a bit outdated. Just like people use MMLU/BigBench to evaluate LLM instead of SST/CB, Such LLM-based ICL approaches need to be validated on MMLU/BigBench to be more convincing.
>
> We appreciate your suggestion of evaluating LLM-based ICL approaches on a more convincing benchmark. We present evaluation results on MMLU below.
>
> | 4-shot ICL         | average | humanities | STEM  | social sciences | other |
> |--------------------|---------|------------|-------|-----------------|-------|
> | LLAMA-65B (Regular) | 63.47   | 68.34      | 54.29 | 72.44           | 64.58 |
> | LLAMA-65B (RdSca)  | 69.36   | 76.18      | 54.67 | 75.35           | 68.20 |
> | LLAMA-30B (Regular) | 58.59   | 60.46      | 48.90 | 67.59           | 62.67 |
> | LLAMA-30B (RdSca)  | 64.75   | 74.32      | 53.00 | 65.66           | 65.00 |
>
> The evaluation results on MMLU clearly demonstrate the advantage of our method over the regular ICL setting. In the revision, we will provide a more comprehensive comparison of different LLMs, numbers of demonstrations, and masking strategies on MMLU/BigBench.
>
> Thanks again for your review! Please let us know if you have any further questions, and we are happy to discuss further.

---

### Official Review · Reviewer_Zcd4 · 2023-08-05

**Soundness:** 3

**Excitement:**

3: Ambivalent: It has merits (e.g., it reports state-of-the-art results, the idea is nice), but there are key weaknesses (e.g., it describes incremental work), and it can significantly benefit from another round of revision. However, I won't object to accepting it if my co-reviewers champion it.

**Paper Topic And Main Contributions:**

- The main motivation of this paper is to overcome the structural limitations of CLMs in ICL.
- To overcome the unidirectional constraints of CLMs, the paper propose RDSCA, which duplicates later demonstrations and concatenates them to the front. Moreover, apply sliding causal attention to prevent repetitive information from occurring.
- The effectiveness of the proposed methodology is demonstrated through various experiments.
- Additionally, the paper performs a thorough analysis of the method's specific elements, such as the importance of the sos token and W-size.

**Questions For The Authors:**

A: Is it possible to provide additional explanations regarding the specific setup of experiments, such as the verbalizer?

**Reasons To Accept:**

- The main motivation of this paper is to overcome the structural limitations of CLMs in ICL.
- To overcome the unidirectional constraints of CLMs, the paper propose RDSCA, which duplicates later demonstrations and concatenates them to the front. Moreover, apply sliding causal attention to prevent repetitive information from occurring.
- The effectiveness of the proposed methodology is demonstrated through various experiments.
- Additionally, the paper performs a thorough analysis of the method's specific elements, such as the importance of the sos token and W-size.

**Reasons To Reject:**

- The coherence between the hypothesis and methodology is positive.
- There is a logical analysis of the hypothesis and its relation to the methodology.
- The well-structured figures significantly aid in understanding the proposed method.

**Reproducibility:**

4: Could mostly reproduce the results, but there may be some variation because of sample variance or minor variations in their interpretation of the protocol or method.

**Reviewer Confidence:**

5: Positive that my evaluation is correct. I read the paper very carefully and I am very familiar with related work.

**Typos Grammar Style And Presentation Improvements:**

- It would be helpful if there were detailed experiment specifics in the appendix. Additionally, providing standard deviations would provide more insight.
- Overall, there is quite a bit of repetition. Reducing redundant content and incorporating more engaging and analytical experimental results would greatly improve the paper.
- Providing a bit more explanation for the last row of T2 would be beneficial. While it is not entirely incomprehensible, it could be made more intuitive for better understanding.

---

> ### Author Rebuttal · Authors · 2023-08-27
>
> We sincerely appreciate your time and dedication in reviewing our paper. In the following, we carefully address each of your concerns point by point.
>
> #### **Answer to the questions**
> > A: Is it possible to provide additional explanations regarding the specific setup of experiments, such as the verbalizer?
>
> Thanks for your suggestion. For the verbalizer, we use semantically-unrelated label words such as 'A', 'B', 'C', and 'D'. For more specific information, please refer to the supplementary materials where we have uploaded the detailed implementation code. We will also include more setup details in the appendix as suggested.
>
> #### **Regarding providing standard deviations would provide more insight**
> > Additionally, providing standard deviations would provide more insight.
>
> Thank you for the great suggestion. We agree that more detailed experiments are helpful and we present the standard deviations for the main experiment below. These results are averaged across six tasks, with the standard deviation computed across 16 runs for each task."
> | std         | 7b   | 13b  | 30b  | 65b  |
> |-------------|------|------|------|------|
> | regular ICL | 1.75 | 0.71 | 2.54 | 3.28 |
> | Rd ICL      | 1.08 | 1.19 | 2.29 | 3.74 |
> | RdSca ICL   | 1.17 | 0.93 | 2.10 | 3.84 |
>
> We will supplement these results in our revision.
>
> #### **Regarding reducing redundant content and providing more analytical experiment**
> > Overall, there is quite a bit of repetition. Reducing redundant content and incorporating more engaging and analytical experimental results would greatly improve the paper.
>
> We appreciate your suggestions and will revise accordingly. Specifically, we will conduct a comprehensive review of our paper to eliminate any redundant content. Furthermore, we plan to incorporate additional analytical experimental results, including evaluations on more advanced benchmarks like MMLU, and an in-depth analysis of sensitivity to demonstration order. If you have any more specific suggestions or recommendations, we would greatly appreciate your input to further enhance the quality of our revision.
>
> #### **Regarding more explanation for the last row of T2**
> > Providing a bit more explanation for the last row of T2 would be beneficial. While it is not entirely incomprehensible, it could be made more intuitive for better understanding.
>
> Thanks for pointing it out. The difference between the last two rows of Table 2 is the number of demonstrations seen by the query. "3-3-3-3-4" means that the query sees all 4 demonstrations while "3-3-3-3-4" means that the query sees the last three. We will provide more explanations and supplement an illustration in Figure 4 to improve the representation in the revision.
>
> Thanks again for your review! Please let us know if you have any further questions, and we are happy to discuss further.

---

### Official Review · Reviewer_evij · 2023-08-09

**Soundness:** 3

**Excitement:**

3: Ambivalent: It has merits (e.g., it reports state-of-the-art results, the idea is nice), but there are key weaknesses (e.g., it describes incremental work), and it can significantly benefit from another round of revision. However, I won't object to accepting it if my co-reviewers champion it.

**Paper Topic And Main Contributions:**

This paper emphasizes the need for proper contextualization of demonstrations in in-context learning, a task that was originally limited by the causal nature of autoregressive language models. To accomplish this goal, the authors propose two methods: "Repeated Demonstrations" and "Sliding Causal Attention." The first approach involves duplicating demonstrations twice, enabling the former demonstrations to reflect the content of the latter ones. Additionally, the second technique modifies attention masks to prevent an over-reliance on duplicated demonstrations. In their research, the authors explored various attention mask variations, providing insight into how these masks inherently function to facilitate in-context learning.

**Reasons To Accept:**

- The proposed techniques are simple, easy to implement, and intuitive.
- Attempted to evaluate different variations of attention masks to investigate the role of the masks for in-context learning.


**Reasons To Reject:**

- The proposed techniques inherently require a significantly higher cost to conduct in-context learning, yet the paper does not address this inefficiency, concentrating only on (potentially marginal) improvements in the final benchmark scores.
- Additionally, the authors focused their testing on the setting of "semantically-unrelated label ICL," which is uncommon in practice. As a result, the practical impact of the proposed method in the real world becomes questionable.
- Finally, the paper would benefit from a clear explanation as to why a specific combination of mask snippets is particularly effective, while most similar ones are not.

**Reproducibility:**

4: Could mostly reproduce the results, but there may be some variation because of sample variance or minor variations in their interpretation of the protocol or method.

**Reviewer Confidence:**

4: Quite sure. I tried to check the important points carefully. It's unlikely, though conceivable, that I missed something that should affect my ratings.

---

> ### Author Rebuttal · Authors · 2023-08-27
>
> We sincerely thank you for your thoughtful feedback! We discuss your raised points as follows:
>
> #### **Regarding extra inference cost**
> > The proposed techniques inherently require a significantly higher cost to conduct in-context learning, yet the paper does not address this inefficiency, concentrating only on (potentially marginal) improvements in the final benchmark scores.
>
> Thank you for the question. We acknowledge that expanding the sequence length may introduce extra computation and memory costs. However, we'd like to highlight how these inefficiencies are mitigated during inference:
> - In real-world, large-scale ICL inference, the representations of demonstrations can be pre-computed and stored offline, leveraging the auto-regressive nature of LLMs. Consequently, when performing ICL for a specific query, there is no need to re-compute the key-value (KV) state for previous tokens. This approach ensures that inference latency remains as low as in regular ICL.
> - From a memory consumption standpoint, 4-shot RdSca is as efficient as 4-shot regular ICL, and significantly more memory-efficient than 7-shot regular ICL. This is due to RdSca's ability to track the status of a maximum of 4 demonstrations for KV-caching, thanks to the sliding window mechanism.
>
> We will expand upon the points mentioned above and provide additional results of training and inference time during revision. Thank you again for the great suggestion.
>
>
> #### **Regarding semantically-unrelated label ICL**
> > Additionally, the authors focused their testing on the setting of "semantically-unrelated label ICL", which is uncommon in practice. As a result, the practical impact of the proposed method in the real world becomes questionable.
>
> Thanks for this insightful feedback. We’d like to provide some clarifications here. Our choice to evaluate our method in the semantically-unrelated label ICL setting was not driven by the performance of our method in the standard ICL setting. Instead, it was motivated by our goal of enhancing the input-label mapping within ICL. Assessing our approach in the semantically-unrelated label ICL setting aligns with this motivation as it relies solely on the input-label mapping to conduct ICL, without the assistance of semantic priors, as explained in section 2.2.
>
> To prevent any potential misunderstandings, we have also conducted experiments in the standard ICL setting on the original MMLU dataset, and the advantages of RdSca over regular ICL are clearly evident.
>
> | 4-shot ICL | average | humanities | STEM  | social sciences | other |
> |-|-|-|-|-|-|
> | LLAMA-65B (Regular) | 63.47   | 68.34      | 54.29 | 72.44           | 64.58 |
> | LLAMA-65B (RdSca)  | 69.36   | 76.18      | 54.67 | 75.35           | 68.20 |
> | LLAMA-30B (Regular) | 58.59   | 60.46      | 48.90 | 67.59           | 62.67 |
> | LLAMA-30B (RdSca)  | 64.75   | 74.32      | 53.00 | 65.66           | 65.00 |
>
> We will supplement the above evaluation results on the standard ICL setting with more specific clarifications in the revision.
>
> #### **Regarding why a specific combination of mask snippets is particularly effective**
> > Finally, the paper would benefit from a clear explanation as to why a specific combination of mask snippets is particularly effective, while most similar ones are not.
>
> We ablate the design of the attention mask in section 4.3, from which we draw some conclusions on why the current design works while its ablations fail. Generally, we attribute the success of sliding causal attention to three factors: preventing information leakage with the sliding window, maintaining the causal constraint of auto-regressively-trained LLMs, and paying additional attention to the <SOS> token while the window slides away. We will provide a clearer explanation of these factors and our findings during revision.
>
> Thanks again for your careful review! Please let us know if you have any further questions, and we are happy to discuss further.

---

### Official Review · Reviewer_n9jP · 2023-08-09

**Typos Grammar Style And Presentation Improvements:** None
**Soundness:** 4

**Excitement:**

4: Strong: This paper deepens the understanding of some phenomenon or lowers the barriers to an existing research direction.

**Missing References:**

None

**Paper Topic And Main Contributions:**

This paper identifies the limitation of causal language modeling in the in-context learning and proposes a novel ICL method, called Repeated Demonstration with Sliding Causal Attention. Experiments on several text classification datasets shows that its method consistently improves the input-label mapping ability of ICL on LLMs of different scales.

**Questions For The Authors:**

I'd like to ask whether this is similar to studying whether the order in which examples are listed in the ICL affects the final result.

**Reasons To Accept:**

1. The limitation of causal language modeling in the ICL proposed in the paper is thought-provoking.
2. Although the repeated demonstration with sliding causal attention is simple, the results show in Figure 2 demostrate its effectiveness.
3. The experiments and demonstrations in this paper are relatively complete and conform to the verification of the conclusions.

**Reasons To Reject:**

The sequence length will be expanded by using repeated demonstration.

**Reproducibility:**

4: Could mostly reproduce the results, but there may be some variation because of sample variance or minor variations in their interpretation of the protocol or method.

**Reviewer Confidence:**

3: Pretty sure, but there's a chance I missed something. Although I have a good feel for this area in general, I did not carefully check the paper's details, e.g., the math, experimental design, or novelty.

---

> ### Author Rebuttal · Authors · 2023-08-27
>
> We sincerely thank you for your time and effort in reviewing our paper.
>
> #### **Regarding expanded sequence length**
> > The sequence length will be expanded by using repeated demonstration.
>
> Thank you for your insightful question. We acknowledge that expanding the sequence length can lead to additional computational and memory expenses, especially during training. However, in the inference phase, the representations of demonstrations can be pre-computed and stored offline, taking advantage of the auto-regressive nature of Large Language Models (LLMs), thereby mitigating the added computational burden. Moreover, our adaptation of the attention mask allows the LLM to focus only on tokens within the sliding window for key-value (KV) caching, rather than retaining information on all previous tokens. This optimization reduces the memory requirements of RdSca. Consequently, the memory consumption of RdSca remains on par with that of a standard ICL setup.
>
> #### **Answer to the questions**
> > I'd like to ask whether this is similar to studying whether the order in which examples are listed in the ICL affects the final result.
>
> Thank you for your great question. Yes, we agree that there is a connection between our study and the study of order sensitivity of ICL demonstrations. Their research has shown that changing the order of demonstrations can greatly impact ICL performance. We hypothesize that this high sensitivity is due to the nonsymmetric property of causal language modeling, which only attends backwardly and causes uneven emphases in all demonstrations, with later demonstrations receiving more information. Therefore, changing the order can cause significant performance perturbations.
>
> From this perspective, our RdSca method should be less sensitive to order, as all demonstrations receive an equal amount of context information. In the revision, we will perform an analysis of the order sensitivity and provide further insights. Thank you again for your insightful suggestions

---

### Meta-Review · Area_Chair_TCjB · 2023-09-19

**Recommendation:** 3

**Metareview:**

This paper argues that causal attention within in-context demos of limits the ICL ability of language models, and the authors propose a simple technique to prepend the demos to the original demos (i.e. demonstration replay) so that each demo example can observe the information of later ones. A sliding window approach is applied to avoid duplicated information in the attention. Experiments on several traditional classification datasets demonstrate the effectiveness of the proposed approach. The pros and cons summarized by the reviewers and the AC are listed below.

### Pros
1. "The limitation of causal language modeling in the ICL proposed in the paper is thought-provoking."
2. "The proposed techniques are simple, easy to implement, and intuitive."
3. "The effectiveness of the proposed methodology is demonstrated through various experiments."

### Cons
> 1. "The sequence length will be expanded by using repeated demonstration."

This point has been raised by multiple reviewers. While the authors have responded in the rebuttal that the KV cache could be pre-computed, storing these precomputed caches still incurs extra cost (maybe relatively minor). I agree with the authors in the rebuttal that the time cost is not increasing with a sliding window approach given that the KV cache is precomputed.

> 2. "On the other hand, 4-shots RdSca ICL cannot outperform regular 7-shots, according to Figure 2, which suggests that using regular ICL might still be the best choice with the same amount of computation budget."

This point is concerning. I do agree with the authors that 4-shot RdSca ICL should be more efficient than regular 7-shots given precomputation. However, a detailed analysis of memory and time efficiency of RdSca ICL is necessary since multiple reviewers are worried about its efficiency -- the 4-shot RdSca ICL is not equivalent on efficiency to regular 4-shots because the first demo of RdSca still attends the other 3 demos while the regular one does not.

> 3. "The selected benchmark seems a bit outdated. Just like people use MMLU/BigBench to evaluate LLM instead of SST/CB, Such LLM-based ICL approaches need to be validated on MMLU/BigBench to be more convincing."

In summary, all the reviewers agree that this paper is sound, but they did not find this paper exciting enough with some concerns as above and results on slightly outdated benchmarks.

---

### Decision · Program_Chairs · 2023-10-07

**Decision:**

Accept-Findings

**Comment:**

This paper argues that causal attention within in-context demos of limits the ICL ability of language models, and the authors propose a simple technique to prepend the demos to the original demos (i.e. demonstration replay) so that each demo example can observe the information of later ones. A sliding window approach is applied to avoid duplicated information in the attention. Experiments on several traditional classification datasets demonstrate the effectiveness of the proposed approach. The pros and cons summarized by the reviewers and the AC are listed below.

### Pros
1. "The limitation of causal language modeling in the ICL proposed in the paper is thought-provoking."
2. "The proposed techniques are simple, easy to implement, and intuitive."
3. "The effectiveness of the proposed methodology is demonstrated through various experiments."

### Cons
> 1. "The sequence length will be expanded by using repeated demonstration."

This point has been raised by multiple reviewers. While the authors have responded in the rebuttal that the KV cache could be pre-computed, storing these precomputed caches still incurs extra cost (maybe relatively minor). I agree with the authors in the rebuttal that the time cost is not increasing with a sliding window approach given that the KV cache is precomputed.

> 2. "On the other hand, 4-shots RdSca ICL cannot outperform regular 7-shots, according to Figure 2, which suggests that using regular ICL might still be the best choice with the same amount of computation budget."

This point is concerning. I do agree with the authors that 4-shot RdSca ICL should be more efficient than regular 7-shots given precomputation. However, a detailed analysis of memory and time efficiency of RdSca ICL is necessary since multiple reviewers are worried about its efficiency -- the 4-shot RdSca ICL is not equivalent on efficiency to regular 4-shots because the first demo of RdSca still attends the other 3 demos while the regular one does not.

> 3. "The selected benchmark seems a bit outdated. Just like people use MMLU/BigBench to evaluate LLM instead of SST/CB, Such LLM-based ICL approaches need to be validated on MMLU/BigBench to be more convincing."

In summary, all the reviewers agree that this paper is sound, but they did not find this paper exciting enough with some concerns as above and results on slightly outdated benchmarks.